# Caffeine and Clinical Outcomes in Premature Neonates

**DOI:** 10.3390/children6110118

**Published:** 2019-10-24

**Authors:** Vasantha H.S. Kumar, Steven E. Lipshultz

**Affiliations:** Department of Pediatrics, University at Buffalo, Buffalo, NY 14203, USA

**Keywords:** caffeine, bronchopulmonary dysplasia, premature infant, newborn, neuroprotection

## Abstract

Caffeine is the most widely used drug by both adults and children worldwide due to its ability to promote alertness and elevate moods. It is effective in the management of apnea of prematurity in premature infants. Caffeine for apnea of prematurity reduces the incidence of bronchopulmonary dysplasia in very-low-birth-weight infants and improves survival without neurodevelopmental disability at 18–21 months. Follow-up studies of the infants in the Caffeine for Apnea of Prematurity trial highlight the long-term safety of caffeine in these infants, especially relating to motor, behavioral, and intelligence skills. However, in animal models, exposure to caffeine during pregnancy and lactation adversely affects neuronal development and adult behavior of their offspring. Prenatal caffeine predisposes to intrauterine growth restriction and small growth for gestational age at birth. However, in-utero exposure to caffeine is also associated with excess growth, obesity, and cardio-metabolic changes in children. Caffeine therapy is a significant advance in newborn care, conferring immediate benefits in preterm neonates. Studies should help define the appropriate therapeutic window for caffeine treatment along with with the mechanisms relating to its beneficial effects on the brain and the lung. The long-term consequences of caffeine in adults born preterm are being studied and may depend on the ability of caffeine to modulate both the expression and the maturation of adenosine receptors in infants treated with caffeine.

## 1. Introduction

Caffeine has been a widespread component of the human diet for thousands of years, given its behavioral and stimulating effects. Friedlieb Ferdinand Runge, considered to be the godfather of caffeine, discovered its chemical structure in 1820 and its relationship to nucleic acid metabolism, especially to purines. Since the 1970s, when the FDA regulated the amount of caffeine in soft drinks over health concerns, interest in caffeine has been high in both the scientific literature and the popular press. 

In the US, caffeine is most often consumed in coffee and soft drinks by people of all ages but also by increasingly younger children. Caffeine consumption among children and adolescents is rapidly increasing, in part because of the introduction of energy drinks containing high doses of caffeine [1]. Up to 75% of young children between 5 to 12 years old [2,3] and up to 15% of 1–2 year olds report consuming some form of caffeine [4].

In healthy adults, caffeine consumption is relatively safe, but for some vulnerable populations such as pregnant women and children, caffeine consumption could be harmful [5]. Risks associated with energy drink use, including those related to sleep loss, might be disproportionately borne by racial minorities and those of lower socioeconomic status [6]. Caffeine is associated with stress, anxiety, and depression, and increases the susceptibility, particularly in children to the potentiating effects of other substances, such as alcohol and illicit drugs [7]. 

Neonates, particularly premature neonates, are increasingly exposed to caffeine because it is the drug of choice in treating apneic events in premature infants. Caffeine has reduced the incidence of bronchopulmonary dysplasia (BPD) without adverse neurodevelopmental outcomes and has become widely used in intensive care units treating such newborns. The review follows the literature search from 2000 to 2018 by Medline relating to effects of caffeine on BPD and neuroprotection in premature infants. 

## 2. Caffeine for Treating Apnea of Prematurity

The inability of the respiratory control mechanisms to respond to changes in PaO_2_ and PaCO_2_ secondary to the immaturity of the central nervous system is thought to be central to the pathophysiology of apnea of prematurity (AOP) [8]. Caffeine and aminophylline act as respiratory stimulants in decreasing the frequency of apnea [9,10,11,12], and have since been confirmed by multiple studies [13,14,15]. In a Cochrane systematic review, methylxanthines (including caffeine and theophylline) reduced both the number of apnea events and the use of mechanical ventilation in premature infants [16]. Caffeine was also associated with better long-term outcomes with minimal toxicity [16,17]. Much better data on its safety and efficacy led to its widespread use globally and the subsequent FDA approval for treating apnea of prematurity in infants. 

Caffeine decreases apnea by stimulating the medullary respiratory centers, increasing the sensitivity to CO_2_ and enhancing diaphragmatic function, which increases minute ventilation and reduces hypoxic respiratory depression [8,18]. Intermittent hypoxemia persists after discontinuation of caffeine and decreases with increasing post-menstrual age (PMA), and prolonged treatment with caffeine reduces these events in premature infants [19]. The effects of apnea on the neurodevelopmental outcome are uncertain at best; however, the severity and duration of hypoxemic events are likely associated with adverse neurodevelopmental outcomes [20].

Caffeine therapy reduced the rates of bronchopulmonary dysplasia (BPD) in very-low-birth-weight infants in the CAP trial (Caffeine for Apnea of Prematurity trial) [21]. In this randomized trial, infants with a birth weight of 500–1250 g received either caffeine or placebo within ten days of birth until AOP no longer occurred. Infants receiving caffeine (loading dose (LD), 20 mg/kg; maintenance dose (MD), 5–10 mg/kg/d) were weaned from positive-pressure ventilation (PPV) one week earlier (median PMA, 31 weeks; IQR, 30–33 weeks) than were infants in the placebo group (median PMA, 32 weeks; IQR, 30–34 weeks). The incidence of BPD was significantly less in the caffeine group in infants who survived to 36 weeks PMA (36% (350/1006) in the caffeine group vs. 47% (447/1000) in the placebo group). Other substantial benefits included substantially less medical therapy for patent ductus arteriosus (PDA) and lower rates of surgical closure of PDA in infants who received caffeine [21]. 

Improved survival without a neurodevelopmental disability was reported for the infants in the CAP trial at 18–21 months [22]. Death or survival with a neurodevelopmental disability occurred in 377 (40.2% (377/937) of infants receiving caffeine compared to (46.2%) (431/932) receiving placebo [22]. On post-hoc analysis, severe retinopathy was less common in infants assigned to the caffeine group (Caffeine Group: 5.1% (49/965) vs. Placebo Group: 7.9% (75/955)) [22]. Fewer days on PPV [21] and a possible decrease in apnea associated with hypoxic events may explain the better neurodevelopmental outcomes and severe retinopathy in the caffeine group. Neonatal caffeine therapy, however, was no longer associated with a significantly improved survival rate without disability in children when assessed at five years of age [23]. The rates of death, motor impairment, behavior problems, deafness, and blindness likewise did not differ significantly between the two groups. However, the secondary analysis demonstrated improvement in gross motor function in the caffeine group at five years of age [23]. 

Studies assessing the functional outcomes from the CAP study up to mid-childhood have had conflicting results. Caffeine did not markedly reduce the short-term, combined rate of academic, motor, or behavioral impairments, but it did reduce the rate of motor impairment at 11 years of age [24]. Also, neonatal caffeine therapy improved visuomotor, visuoperceptual, and visuospatial abilities at 11 years of age [25]. General intelligence, attention, and behavior were not adversely affected by caffeine, highlighting the long-term safety of caffeine therapy for apnea of prematurity in very low birth weight neonates [25]. 

Ex-preterm infants, regardless of neonatal caffeine therapy, are at risk for obstructive sleep apnea and periodic limb movements in later childhood [26]. However, caffeine had no long-term effects on the prevalence or duration of sleep apnea during childhood [26]. Earlier extubations and less severe BPD led to improvements in expiratory flow rates in mid-childhood in these infants [27]. The CAP trial and other studies have established the relative safety and efficacy of caffeine therapy in the treatment of AOP and BPD has become one of the most cost-effective pharmacotherapy in neonatal intensive care (NICU) [28]. 

## 3. Pharmacology of Caffeine in Premature Neonates

Caffeine (1,3,7-trimethylxanthine, C_8_H_10_N_4_O_2_) is a plant alkaloid structurally similar to adenosine. Micromolar concentrations of caffeine are competitive antagonists of the G-protein coupled adenosine receptors, particularly of A_1_ and A_2a_ receptors [29]. In humans, 99% of the consumed caffeine is absorbed within 45 min of ingestion, mostly in the small intestine [30]. Caffeine binds reversibly to plasma proteins, and protein-bound caffeine accounts for 10%–30% of the total plasma pool. Most premature infants have plasma concentrations of the caffeine between 5–20 µg/mL after receiving standard doses (5–10 mg/kg/day) of caffeine [31]. 

Caffeine is hydrophilic, diffuses freely into intracellular tissue water in all biologic fluids with a high volume of distribution of 0.8–0.9 L/kg in infants [32]. The distribution is greatly affected by postnatal age and current weight [33]. Caffeine is also sufficiently lipophilic to pass through all biological membranes and readily crosses the blood-brain barrier with similar concentrations in serum and cerebrospinal fluid [34]. Caffeine crosses the placenta by passive diffusion and is widely distributed in fetal tissues [35], and is readily excreted in breast milk, [36] resulting in a high milk-to-serum concentration ratio. 

Caffeine is metabolized almost exclusively in the microsomal enzyme system of the liver, predominantly by the enzyme CYP1A2, which is essentially a conversion to N-3 (paraxanthine), N-7 (theophylline), and N-1 (theobromine) demethylation [37]. The predominant metabolic pathway in adults is through N-3 demethylation to paraxanthine (70%–80%) [37] however, in preterm infants N-7 demethylation is the main metabolic pathway that matures at about four months of age [38]. The demethylation process depends on postnatal age, regardless of gestational age (GA) or birth weight; however, infant girls metabolize caffeine at a higher rate than do infant boys [38]. The serum half-life in infants and premature neonates varies from 40–230 h and has decreasing half-lives with advancing PMA until 60 weeks of age [39]. 

In full-term and premature infants, caffeine clearance is a function of gestational and postnatal age; caffeine clearance is slower at birth and increases with age with increasing glomerular filtration rate [40]. Immaturity of the microsomal enzyme systems in the liver results in most of the caffeine being excreted unchanged in the urine in neonates, [39] in contrast to being almost completely metabolized in adults [41]. 

## 4. Caffeine’s Mechanism of Action

Caffeine is structurally similar to adenosine, a neuromodulator whose metabolism depends on ATP synthesis, release, and breakdown. The physiological functions of adenosine are mediated through G-protein coupled receptors. Of the four adenosine receptors, A_1_, A_2a_, A_2b_, and A_3_, micromolar concentrations of caffeine block the A_1_ and A_2a_ receptors (Figure 1) [29]. Adenosine receptors, specifically A_2a_ and A_2b,_, are in moderate concentrations in the lung; whereas the brain has high concentrations of A_1_ and A_2a_ receptors [42]. Inhibitory A_1_ receptors are ubiquitous in the brain with the highest concentrations in the hippocampus and the neocortex [43]. Adenosine exerts its modulatory role through activation of A_1_ receptors by suppressing synaptic activity [44] and inhibiting neurotransmitter release [45]. Excitatory A_2A_ receptors are mainly in the striatum [46], with an ability to control synaptic plasticity and neurodegeneration [47]. Caffeine affects the release of neurotransmitters, such as norepinephrine, dopamine, acetylcholine, serotonin, glutamate, gamma-aminobutyric acid (GABA) and neuropeptides by blocking the adenosine inhibitory signals through its receptors [48,49], that have significant effects on the perception of alertness and wakefulness besides influencing behavioral and cognitive performance [50,51] (Figure 1). The effects of caffeine are dependent on the dose administered. At low doses caffeine antagonizes the action of adenosine at A_1_, A_2a_, and A_2b_ receptors; however the effects are not only qualitatively different but may also have a different mechanism of action at high doses [52]. Low-dose caffeine result in a plasma caffeine concentrations of ~20 mM, often sufficient to generate mild neurological stimulatory actions [53]. However, higher caffeine levels (>75 mM) are required to evoke adverse effects, which can become fatal at levels of >400 mM [54]. Hence, care must be taken to take into account the dose of caffeine in epidemiological and experimental data while drawing conclusions on the pathophysiology of adenosine [52]. 

Caffeine mobilizes intracellular calcium from the endoplasmic reticulum through activation of the ryanodine receptor (RyR2) channels [55] (Figure 1). Caffeine induces Ca^2+^ release by reducing the threshold for luminal Ca^2+^ activation of the RyR2 channel [55]. However, cytosolic calcium increases RyR2 caffeine affinity, suggesting that activation of the RyR2 channel depends on free Ca^2+^ on both sides of the channel [56]. Low-dose caffeine (0.15 mM) increases both the calcium spark frequency and the RyR channel opening frequency [56]. Activation of ryanodine receptor channel stimulates excitation-contraction coupling and also augments the capacity of neurons to release neurotransmitters and to improve neuronal survival [57]. Intracellular calcium also increases the production of nitric oxide by activating endothelial nitric oxide synthase [58,59]. The associated changes in the neuromuscular junction and the skeletal muscle enhance physical performance [60]. 

Caffeine, by non-selectively inhibiting phosphodiesterase, increases intracellular concentrations of cAMP in skeletal muscle and adipose tissue [61] (Figure 1). Indeed, caffeine was the first phosphodiesterase inhibitor identified, and the first enzymatic effect of caffeine to be defined. An increase in cAMP leads to a rise in blood catecholamine and release of free fatty acid and glycerol from the adipose tissue by activating lipases [62]. Caffeine inhibition of phosphodiesterase requires higher concentrations (100–1000 µM) than that of its interaction with the adenosine receptors (10–100 µM) [63]. Adenosine receptor blockade results in positive inotropy and chronotrophy in the heart from heightened β1-receptor activity; with an increase in AV conduction velocities and heart rate predisposing to tachycardia and arrhythmias.

Electron spin resonance spectroscopic measurements indicate that a caffeine-derived oxygen-centered radical is formed when caffeine reacts with a hydroxyl radical (•OH), which explains the anticarcinogenic properties of caffeine and related methylxanthine compounds [64]. Caffeine inhibits lipid peroxidation and oxidative damage from reactive oxygen species [65]. In newborn rats exposed to hyperoxia, pretreatment with caffeine reduced oxidative stress, promoted anti-oxidant responses, down-regulated pro-inflammatory cytokines, modulated the expression of redox-sensitive transcription factor, reduced pro-apoptotic effectors, and diminished extracellular matrix generation [66]. Along with its ability to scavenge free radicals, caffeine is uniquely positioned to protect the developing brain, and some of these actions are mediated by the adenosine receptors [67,68]. 

Maturation of adenosine receptors A1 and A2 in tissues is developmentally regulated. 

## 5. Caffeine, Bronchopulmonary Dysplasia, and Lung Protection

Historically, methylxanthines are administered before extubation to increase the chance of successful weaning from mechanical ventilation (MV). Caffeine facilitates successful extubations by stimulating breathing in premature infants [69], and short-term benefits of successful extubation are more likely with standard doses of caffeine (20 mg/kg/d) than with low doses (5 mg/kg/d) [70]. Higher doses of caffeine (LD, 40 mg/kg; MD, 20 mg/kg/d) decrease the chances of extubation failure in infants on MV [71]; however, very high doses of caffeine (LD, 80 mg/kg) are associated with a higher risk of cerebellar hemorrhage [72]. Caffeine administered to facilitate extubation also improves AOP over the short-term in premature infants [73,74]. 

High-dose caffeine (LD, >20 mg/kg/d; MD, >10 mg/kg/d) prevented chronic lung disease with fewer extubation failures, apnea and shorter duration of MV in these infants [75]. Neonates receiving caffeine therapy within the first three days had better outcomes than neonates receiving caffeine three days after birth, including death or BPD and PDA requiring treatment [76]. In another study, caffeine intake which started in the first two days after birth was associated with better survival without BPD than when administered 3–10 days after birth [77], suggesting that the earlier the treatment, the better its protective effects on the lung. Higher maintenance doses of caffeine (10–20 mg/kg/d) also decreased the incidence of BPD in a meta-analysis [73]. However, except for the CAP trial, no other randomized studies have addressed the relationship between caffeine and BPD in premature infants. Evidence from retrospective studies and meta-analyses indicate that earlier administration of caffeine and higher doses may have substantial benefits on the lung. The benefits of earlier caffeine on BPD may be related to the unique physiology of the fetus and the premature neonate modulated by the ductus arteriosus. Caffeine’s ability to decrease medical treatment for PDA suggests that it favorably alters the hemodynamics of PDA and ductal closure by attenuating the fluctuations in systemic blood pressures, may offer substantial benefits to the cerebral circulation. However, in preterm infants with PDA, caffeine increased flow through the ductus arteriosus before constricting it 4 h after the loading dose [78]. 

Studies have addressed the molecular mechanisms by which caffeine protects the lungs, especially hyperoxia in animal models, but have had conflicting results. Caffeine reduces inflammation [79,80], attenuates endoplasmic stress, [81] and prevents hyperoxia-induced functional and structural lung damage [79,81] in animal models. Studies in murine models of hyperoxia-induced lung injury have reported potentially adverse effects of caffeine on alveolar development, with increased inflammation and worsening alveolar hypoplasia, along with increased cellular apoptosis and decreased expression of A_2a_ receptors [82]. Caffeine modulates TGF-β signaling but does not attenuate the blunted alveolarization in hyperoxia-induced lung injury [83]. We have reported that administering caffeine to newborn mice improves alveolarization in 12-week old adult mice [84] and most likely, its effects are mediated by regulation of angiogenesis in hyperoxia-induced lung injury [84]. 

Prolonged caffeine treatment (0.1 g/L–0.25 g/L over 2 weeks, or acute caffeine treatment at high doses (50 mg/kg intraperitoneal), significantly attenuated lung edema, hemorrhage, and neutrophil recruitment, as well as the inflammatory cytokine response in both the wild type and A_2a_R knockout (KO) mice [85]. This attenuation was accompanied by an increase in cAMP levels and up-regulation of A_2b_R mRNAs in the lungs. In contrast, low-dose caffeine (5–15 mg/kg intraperitoneal) administered as an acute treatment before acute lung injury worsened inflammation and lung damage in wild-type mice with a decrease in cAMP but not in A_2a_R KO mice. These results indicate that caffeine either enhances lung damage by antagonizing A_2a_ receptors or protects against lung damage through A_2a_R-independent mechanisms, depending on the timing of exposure (chronic vs. acute) and the dose of administration (low vs. high) [85].

Differing doses of caffeine for varying duration have reported opposite effects on the lung, with some reporting benefits and others summarizing the damage in animal studies. The results are equivocal, partly because the doses of caffeine administered (4.8–1.2 mg/kg/d) are lower in human-equivalent doses and are of varying duration (1–14 d) in these studies. Collectively, these studies do not identify a predominant mechanism of action of caffeine in the lung. However, its effect on adenosine receptors at low doses and on inhibiting phosphodiesterase at high doses, along with the duration administered and the unique pharmacodynamics of caffeine in neonates may help determine its clinical effects on the lung. 

## 6. Caffeine and Neuroprotection

Prolonged mechanical ventilation by itself is a strong risk factor for poor neurodevelopmental outcomes at 18 months of age [86]. Infants in CAP trial on caffeine and had fewer days of PPV had less motor impairment [24]. More recently, starting caffeine within two days of birth dramatically reduces the odds of neurodevelopmental impairment at ages 18–24 months in infants <29 weeks gestation [87]. Among extremely preterm infants who survived to 36 weeks PMA, prolonged hypoxemic episodes during the first 2–3 months after birth are associated with adverse 18-month outcomes [20]. Extended caffeine therapy reduces recurrent hypoxemic events in these premature infants [19]. Caffeine’s favorable effect on cardiorespiratory physiology in stabilizing systemic and cerebral hemodynamics and its capacity to mitigate hypoxic respiratory depression may play a part in neuroprotection. 

The effects of caffeine administered during pregnancy and in the immediate newborn period have mostly come from animal studies, and the results are equivocal at best. In mouse dams, treated with either caffeine or other A_2A_R antagonists during pregnancy and lactation, migration and insertion of GABA neurons into the hippocampal circuitry in their offspring were delayed during the first week after birth [88]. Adult offspring of these mouse dams have fewer hippocampal GABA neurons, some cognitive deficits, increased neuronal network excitability, and susceptibility to seizures in response to a seizure-inducing agent. These effects suggest that exposure to caffeine during pregnancy and lactation in rodents may have adverse effects on the neural development of their offspring [88]. 

Animal models of fetal drug exposure to cocaine, alcohol, or caffeine consistently reveal impaired GABA neuron development [89]. Caffeine may also trigger seizures in susceptible individuals by increasing intracellular calcium and protein kinase activity in neurons [90] and decrease the antiepileptic potency of certain drugs, especially topiramate [91]. Caffeine overdose (36 mg/kg to 136 mg/kg) and perinatal asphyxia may precipitate or increase seizure activity in term neonates [92]. Early high-dose caffeine (80 mg/kg) is associated with an increase in seizure incidence and seizure burden in premature infants ≤30 weeks GA [93]. 

Daily high-dose caffeine does not adversely affect the developing white matter in ovine fetuses [94]. However, high doses transiently decrease myelin formation in newborn mice [95] and increase neuronal cell death when administered with or without morphine in rats [96], suggesting that the effects of caffeine are not only dose-specific but also species-specific. Physiologically relevant doses (20–30 mg/k/d) of caffeine can markedly depress hippocampal neurogenesis in adult mice [97]. Higher doses of caffeine (60 mg/k/d) despite increasing proliferation does not improve long-term neurogenesis and in fact, may have decreased survival than control neurons [97]. Alterations in astrocyte densities in the brain during development, from A_2A_ receptor blockade by postnatal caffeine, can have long-term consequences on the brain function [98]. In our study, no adverse effects on learning and memory were noted in adult mice after postnatal caffeine in newborn mice [99]. However, the beneficial effects on spatial learning may be gender specific, with males performing better than female adult mice [99].

An understanding of the maturation and density of adenosine receptors through development is crucial to the understanding of the long-term effects of caffeine in premature infants. Postnatal changes in expression of adenosine receptors have been described in the rodent brain [100,101] A_1_ receptor expression increases markedly with progressive maturation [43], with increasing density and functional maturity from postnatal days 10–15 days up to days 25–40 [102,103]. Levels on P25 and older rats are significantly lower than levels in P10–P12 brains [103]. A_1_ receptors are widely distributed at birth (around 10% of adult levels), increasing gradually until adulthood, and peaking during the second week of postnatal life [43,102,103]. Moreover, the affinity and binding capacity of A_1_ receptor decrease in the hippocampus upon aging [104]. A_1_ receptors are widely present in different tissues including brain endothelial cells with consequences, on cell growth and angiogenesis during postnatal development [103]. ATP-derived adenosine preferentially activates pro-convulsive A_2_ receptors than inhibitory A_1_ receptors [105]. The developmental profile of A_2A_ receptors reveal low A_2A_ receptor density at birth and a gradual increase postnatally [103]. The density of A_1_ receptor binding sites decreases in the hippocampus, in contrast to the increased number of A_2A_ receptor binding sites in aged rats versus young rats [104], suggesting age related symmetry between inhibitory A_1_- and excitatory A_2A_-mediated actions. 

The data from both the animal and human studies described above are not consistent with those from infants in the CAP trial. However, several factors have to be considered when translating findings from animal models to human populations [89]. Despite the differences in brain development in rodents and primates, animal models are nevertheless relevant to human brain development. Traditional models of developmental brain injury have utilized rodents at postnatal days 7–10 as being roughly equivalent to a term infant [106]. The timing of neurogenesis, oligodendrocyte maturation, gliogenesis, and synaptogenesis coincide with developmentally regulated molecular and biochemical changes. Even though the time scale is considerably different, the sequence of key events in brain maturation are largely consistent between humans and rodents. For a premature infant between 23–32 weeks of age, the GA corresponds to postnatal day 1–3; developmentally corresponding to oligodendrocyte maturation, immune system development and establishment of the blood-brain barrier [106]. For a term infant at 36–40 weeks of age, the GA corresponds to postnatal day 7–10 in rodents; as does the age of peak brain growth spurt, gliogenesis, and increased axonal and dendritic density [106]. In rodents, the term ‘adolescence’ is used to cover the whole period from weaning at postnatal day 21 to adulthood at postnatal day 60 [107]. Expression of adenosine receptors in general and the balance of inhibitory A_1_ and excitatory A_2A_ receptors in particular in the early postnatal period and in adolescence may have implications on the long-term effects of caffeine in premature neonates. However, this information is not known at present and needs further investigation. 

Preterm infants (≤30 weeks gestation) randomly assigned to receive higher loading-dose of caffeine (80 mg/kg vs. 20 mg/kg) had a higher incidence of cerebellar hemorrhage with early motor alterations at term equivalent age [72]. However, MRI measurements of brain metrics at term age and developmental assessment at two years were not different between the groups [72]. The high doses of caffeine associated with adverse effects on brain development in both animal and human studies are unlikely to be administered to infants; however, the standard dosing (LD, 20 mg/kg; MD, 5–10 mg/kg) needs to be studied further in premature infants. Basic and translational research is required on GABA neurons and adenosine receptor densities across brain regions during development to maximize the therapeutic efficacy of caffeine across different doses. 

Caffeine may have different effects when administered prenatally rather than postnatally; the transitional changes at birth concerning hormones and neurotransmitters and their impact on adenosine receptor development need to be studied. Studies of brain development in fetuses and premature infants at term equivalent should incorporate non-invasive imaging, such as magnetic resonance imaging (MRI) and positron emission tomography (PET). Progressive myelination and a relative decrease in water content with development are observable by MRI. When coupled with an assessment of analytical functions and receptor expression at the cellular level by PET, a clear picture may emerge regarding the effects of caffeine on the developing brain. We need to determine whether the beneficial effects of postnatal caffeine administration are a result of structural or physiological alterations or a combination of events (Figure 2). 

Newborn physiology is unique in terms of the presence of PDA and the immature cerebral autoregulation in the developing brain. The benefits of caffeine when administered within 48–72 h after birth [76,77] and the reduced need for medical and surgical treatment of PDA [21] indicate that caffeine may stabilize the fluctuations in systemic blood pressure, and hence cerebral blood flow, offering neurologic benefits to premature infants. Assessing cerebral blood flow, cerebral oxygen saturation, and cerebral autoregulation in these infants may provide additional clues to the physiologic benefits of caffeine. Studies need to link cardiovascular benefits to cerebral physiology and structural alterations in the brain to define the neuroprotective effects of caffeine.

## 7. The Long-Term Effects of Neonatal Caffeine Exposure

Adverse fetal programming from in-utero exposure to caffeine has been associated with an increased risk of childhood obesity [111]. The observation is further supported by a dose-response relationship between the amount of maternal caffeine intake during pregnancy and further increased risk of obesity in offspring up to 15 years of age [111]. This relationship is stronger for persistent obesity and more common in girls [111]. In a nationwide cohort of pregnant women, any caffeine consumption during pregnancy modified the overall infant growth trajectory from birth to 8 years of age by increasing the risk of excess infant growth and childhood overweight [112]. 

Maternal caffeine exposure is also associated with an increased risk of miscarriage [113], low birth weight, fetal growth restriction, and being small-for-gestational-age at birth [114,115]. Prenatal exposure to caffeine predisposes the offspring to excess growth and cardiometabolic changes by altering the hypothalamic-pituitary-adrenal axis [116,117], regulating adenosine and its receptors that modulate development, [118,119] and by the placental expression and transport of leptins, which are essential for regulating appetite [120]. Studies [112] reporting excess infant weight and growth velocity are based on self-reported dietary data with a median maternal caffeine intake of less than 60 mg/d (25th–75th Centile: 23–120 mg/day; about 1–2 mg/kg/d) under normal circumstances; this is in contrast to the markedly higher doses administered to premature infants and its possible effects on the developing brain and other organ systems. 

The effects of neonatal caffeine exposure on behavior in children, particularly in adolescents, on cardio-metabolic profiles, systemic hypertension, or stress are being studied. Neonatal caffeine exposure can increase systemic blood pressure in adult mice with an increase in vessel reactivity, especially in males [121] (under publication), suggesting that programming at the hypothalamic-pituitary level may be sex-specific. In humans, the risk of adult-onset diseases, such as diabetes, cardio-metabolic disease, and psychosocial morbidities in adults born preterm with caffeine exposure at birth is unknown.

The effects of co-administration of substances of abuse, such as nicotine or additional caffeine, in adolescents and young adults of ex-preterm infants are unknown. Numerous studies have reported adverse effects of caffeine in adolescents. Caffeine consumption in adolescence may predispose to psychiatric disorders, including anxiety-related disorders, possibly from dysregulating the neuroendocrine stress response system [122]. Caffeinated alcoholic drinks increase the risk of alcohol-related consequences [123] and affect anxiety, recognition memory, [124] sleep duration, and daytime behaviors [125]. The effects of addictions to sugar and caffeine in preterm infants prone to overweight and caffeine tolerance from neonatal exposure also need to be determined. The carcinogenicity of drinking coffee is being intensly debated over the past several decades. In 2016, the Working Group of the International Agency for Research on Cancer (IARC), has determined that there is inadequate evidence in humans for carcinogenicity of coffee drinking [126]. There is no consistent evidence of an association between bladder cancer and coffee drinking as studies have been confounded by smoking and occupational exposures in men. In fact, a decreasing dose-response relationship between coffee consumption and the risk for myelodysplastic syndrome (MDS) among men was noted in a recent study [127]. Stratified analysis of the relative risks for acute myeloid leukemia and MDS of coffee drinkers relative to non-drinkers demonstrated the confounding effects of smoking in coffee drinkers [127]. However, caffeine administered to premature infants during the period of rapid development is a relatively new area of interest and the association needs investigation in children. 

Whether caffeine or illicit drugs can alter cortical neuronal migration, specifically GABA inhibitory neurons during brain development in humans, is still uncertain. What we need are definitive scientific data from human studies on what, if any, adverse effects on brain development and adult behaviors are linked to caffeine use during pregnancy or exposures to caffeine in premature neonates. These data are especially crucial because animal studies have convincingly demonstrated adverse effects on hippocampal GABA neurons in the offspring of dams treated with caffeine. The effects of caffeine on brain development could have long-term consequences on neurobehavioral and neuropsychiatric disorders in adults. Caffeine effects across development and through aging are variable, with beneficial benefits reported in premature infants and adverse effects predominantly in pregnancy and childhood (Figure 2).

Advances in perinatal care, especially regarding surfactant administration in premature infants and the use of inhaled nitric oxide for persistent pulmonary of the newborn in term infants are classic examples in which several randomized trials have found beneficial effects in both animal models and newborns [128,129,130]. With the ubiquitous presence of adenosine receptors in the human body, receptor modulation by caffeine affects multiple organ systems. With the extensive use of caffeine to treat AOP and in the prevention of BPD in premature infants, studies should address the long-term safety of caffeine on various organ systems as these infants grow into adults. 

## 8. Conclusions

The evidence supports reducing caffeine intake during pregnancy and adolescence and moderate consumption in adults. However, the evidence supports caffeine administration to treat AOP and to improve lung and brain development in premature infants. A therapeutic window for caffeine in infants between 24 weeks and 44 weeks gestation might be determined by fetal development, sex, oxygen exposures, and other factors. Defining this therapeutic window and the factors associated with favorable outcomes in these infants is the next challenge. 

## Figures and Tables

**Figure 1 children-06-00118-f001:**
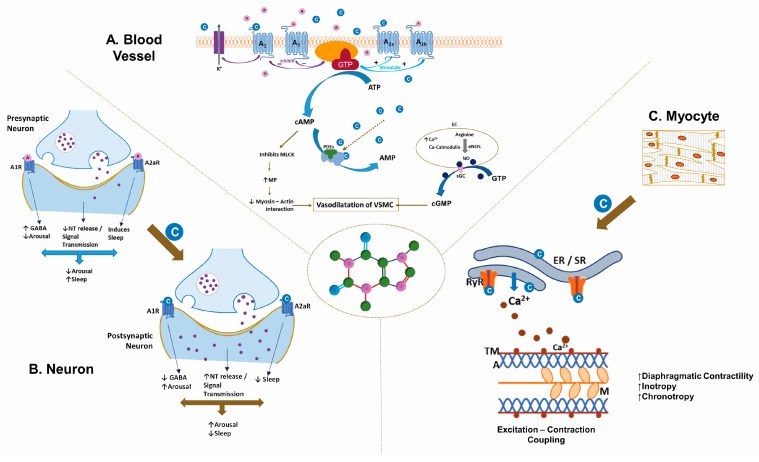
Mechanism of action of caffeine (C_8_H_10_N_4_O_2_). The illustration depicts the distinct actions of caffeine on (**A**). Blood Vessel, (**B**). Neurons, and (**C**). Myocytes. (**A**). In the vasculature of smooth muscle, caffeine inhibits myosin light-chain kinase via the cAMP pathway and stimulates myosin light-chain phosphatase, inhibiting the myosin-actin interaction. The A_1_/A_3_ receptors inhibit cAMP production, whereas A_2a_/A_2b_ receptors stimulate cAMP production by adenylate cyclase. Caffeine increases endothelial cell (EC) Ca^2+^, stimulating endothelial nitric oxide synthase to produce nitric oxide, dilating the vascular smooth muscle cells through the cGMP pathway. (**B**). Caffeine antagonizes the actions of adenosine at the adenosine receptor, specifically at the A_1_/A_2a_ receptors, resulting in increased wakefulness coupled with altered activity of neurotransmitters such as dopamine, epinephrine, norepinephrine, and serotonin in the nervous system. (**C**). Caffeine mobilizes intracellular calcium by activating the ryanodine receptor on the sarcoplasmic reticulum of skeletal and smooth muscles, facilitating excitation-contraction coupling. A_1_, A_3_, A_2a_, and A_2b_, adenosine receptors; cAMP-cyclic Adenosine Monophosphate; AC-adenylate cyclase; PDEs-Phosphodiesterase; AMP-adenosine monophosphate; GTP-Guanosine triphosphate; cGMP-cyclic Guanosine Monophosphate; sGC-soluble guanylate cyclase; C-Caffeine molecules; A-Adenosine molecules; ER/SR-Endoplasmic-sarcoplasmic reticulum; TM-tropomyosin; A-actin; M-myosin; ECC-excitation-contraction coupling; NT-neurotransmitters; GABA-gamma-aminobutyric acid (copyright:Vasantha H.S. Kumar, MD).

**Figure 2 children-06-00118-f002:**
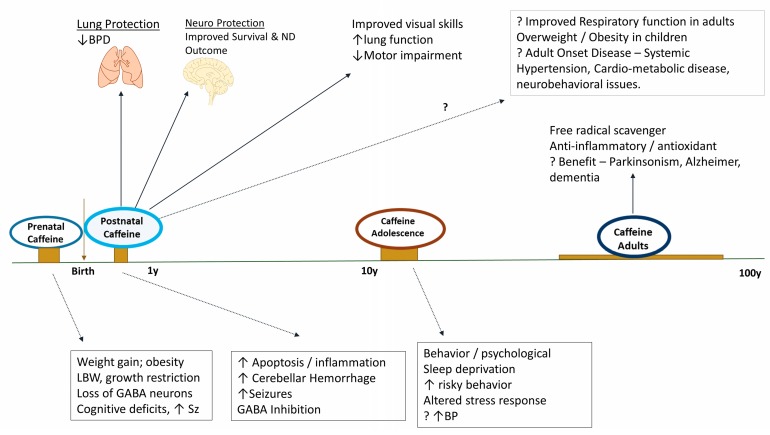
The spectrum of effects when caffeine is consumed prenatally, postnatally, and during adolescence. Caffeine consumed during pregnancy and adolescence may have adverse effects. However, in premature infants, caffeine is remarkably safe and has substantial benefits on the lungs and the brain. Caffeine reduces bronchopulmonary dysplasia, improves survival and neurodevelopmental outcomes at 18 months in premature infants ≤1250 g at birth. Infants should be monitored for cardio-metabolic, neurobehavioral, and other adult-oriented disorders as the long-term effects of caffeine are not known. Caffeine consumption offers substantial benefits in adults [108,109,110].

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
