# Peer review of "Caffeine and Clinical Outcomes in Premature Neonates"

_children, 2019, doi:10.3390/children6110118_

Round 1

Reviewer 1 Report

The present review is considering the mid- and long-term consequences of the exposure of premature infants to caffeine used in the treatment of apnea. The authors considered both human and animal studies and have written an excellent and detailed review on the topic. The coverage of the existing literature in the field is extensive and good. I only have a few comments.

In part 4 concerning the mechanisms of action of caffeine, several references used would need to be updated. Refs 43 and 44 use very high concentrations of caffeine that are not relevant to the human situation. Ref 45 and 46 are totally outdated. For more recent information on these topics the authors should consider reading articles published by M Morelli, JA Ribeiro, R Cunha who have published several recent reviews on caffeine and adenosine. The following article is of particular interest and could be used in this review: Fredholm BB, Yang J, Wang Y. Low, but not high, dose caffeine is a readily available probe for adenosine actions. Mol Aspects Med. 2017;55:20-25. doi: 10.1016/j.mam.2016.11.011. Ref 45 should be replaced by a more recent one since many papers on mood and positive central effects were reported since 1982. A possibility is Fredholm et al. 1999, Pharmacol Rev but there are many even much more recent ones available. The part on calcium and ryanodine receptors should be rewritten in light of the caffeine dosages used in humans. In fact, in this whole part, the range of concentrations at which the different mechanisms of action of caffeine might be involved is missing along with their relevance to caffeine’s actions in vivo linked to the doses administered. This was mentioned only for PDE.

In this part the issue of age is not addressed either. There are data in the literature on the maturation of adenosine receptors in the brain at least (see for example: Doriat JF, Humbert AC, Daval JL. Brain maturation of high-affinity adenosine A2 receptors and their coupling to G-proteins. Brain Res Dev Brain Res. 1996;93(1-2):1-9.). This point needs to be considered in a review dealing with immaturity. Indeed, how is caffeine acting in the premature infant since none of the systems by which caffeine is acting is fully developed. We probably do not have an answer to this question but it needs at least to be raised here.

The second issue which is not discussed here is the correspondence between the level of maturity in humans at given stages and the corresponding one in rodents. Such a scale can be found for the various steps of brain maturation. I am not working on the lung so I do not know what is available in that area. The authors need to address this point because, as raised earlier, we need to understand in which respect animal experimentation is comparable to the human prematurity situation and looking more closely at those points would help having a different outlook at literature and interpretation of the data as well as giving new inspiration for further studies on this topic.

Finally, in the description of the consequences of early exposure of the human fetus to caffeine, the authors do not evoke at all one point that has been raised by several papers which is the risk for the child exposed in utero to caffeine to develop leukemia. The recent conclusions of IARC were that we do not have enough evidence at this point and the data are not considered solid enough because everything is based on retrospective recollection of the quantities of caffeine consumed during pregnancy some 6-10 years later (Drinking Coffee, Mate, and Very Hot Beverages. IARC Working Group on the Evaluation of Carcinogenic Risk to Humans. Lyon (FR): International Agency for Research on Cancer; 2018). However, in the list of comorbidities cited by the authors, this one could be added although we still have no certitude and no prospective data but this issue is raised regularly.

Author Response

The present review is considering the mid- and long-term consequences of the exposure of premature infants to caffeine used in the treatment of apnea. The authors considered both human and animal studies and have written an excellent and detailed review on the topic. The coverage of the existing literature in the field is extensive and good. I only have a few comments.

In part 4 concerning the mechanisms of action of caffeine, several references used would need to be updated. Refs 43 and 44 use very high concentrations of caffeine that are not relevant to the human situation. Ref 45 and 46 are totally outdated. For more recent information on these topics the authors should consider reading articles published by M Morelli, JA Ribeiro, R Cunha who have published several recent reviews on caffeine and adenosine. The following article is of particular interest and could be used in this review: Fredholm BB, Yang J, Wang Y. Low, but not high, dose caffeine is a readily available probe for adenosine actions. Mol Aspects Med. 2017;55:20-25. doi: 10.1016/j.mam.2016.11.011. Ref 45 should be replaced by a more recent one since many papers on mood and positive central effects were reported since 1982. A possibility is Fredholm et al. 1999, Pharmacol Rev but there are many even much more recent ones available. The part on calcium and ryanodine receptors should be rewritten in light of the caffeine dosages used in humans. In fact, in this whole part, the range of concentrations at which the different mechanisms of action of caffeine might be involved is missing along with their relevance to caffeine’s actions in vivo linked to the doses administered. This was mentioned only for PDE.

In this part the issue of age is not addressed either. There are data in the literature on the maturation of adenosine receptors in the brain at least (see for example: Doriat JF, Humbert AC, Daval JL. Brain maturation of high-affinity adenosine A2 receptors and their coupling to G-proteins. Brain Res Dev Brain Res. 1996;93(1-2):1-9.). This point needs to be considered in a review dealing with immaturity. Indeed, how is caffeine acting in the premature infant since none of the systems by which caffeine is acting is fully developed. We probably do not have an answer to this question but it needs at least to be raised here.

The second issue which is not discussed here is the correspondence between the level of maturity in humans at given stages and the corresponding one in rodents. Such a scale can be found for the various steps of brain maturation. I am not working on the lung so I do not know what is available in that area. The authors need to address this point because, as raised earlier, we need to understand in which respect animal experimentation is comparable to the human prematurity situation and looking more closely at those points would help having a different outlook at literature and interpretation of the data as well as giving new inspiration for further studies on this topic.

Finally, in the description of the consequences of early exposure of the human fetus to caffeine, the authors do not evoke at all one point that has been raised by several papers which is the risk for the child exposed in utero to caffeine to develop leukemia. The recent conclusions of IARC were that we do not have enough evidence at this point and the data are not considered solid enough because everything is based on retrospective recollection of the quantities of caffeine consumed during pregnancy some 6-10 years later (Drinking Coffee, Mate, and Very Hot Beverages. IARC Working Group on the Evaluation of Carcinogenic Risk to Humans. Lyon (FR): International Agency for Research on Cancer; 2018). However, in the list of comorbidities cited by the authors, this one could be added although we still have no certitude and no prospective data but this issue is raised regularly.

The authors thank the reviewer for excellent comments and suggestions.

We have rewritten the section on Caffeine mechanism of action. This is in more detail and we have incorporated the references suggested by the reviewer (a copy of track changes of the manuscript is attached).

We have written an extensive paragraph on adenosine receptor expression and maturation through development as this is relevant to the topic, especially in premature neonates.

We have compared mice and human development through development to give the reader the perspective on some of the comparative data available currently.

The authors have incorporated some of the studies related to caffeine and cancer.

The changes related to adenosine receptor expression through development, comparative development status in mice and humans and literature related to cancer are in the section: Long-term effects of caffeine.

Again, we thank the reviewer for suggestions. These changes have made the review stronger.

Reviewer 2 Report

Drs Kumar and Lipshultz have reviewed the use of caffeine and clinical outcomes in premature neonates.1. The literature review methodology should be described to demonstrate that the results are comprehensive and generalisable.
2. In the abstract – caffeine is described as a psychoactive drug – perhaps not the best description in the first few sentences
3. It would be important to describe the most recent results of the CAP trial
4. The abstract should be modified to take into consideration the pre-natal animal results
5. The reference list is incomplete

Author Response

Reviewer 2.

The literature review methodology should be described to demonstrate that the results are comprehensive and generalisable.

We have included literature related to lung and neuroprotection in premature protection from 2000 to 2018 by Medline search. There is extensive literature on caffeine in general and this review is specific to caffeine in premature neonates related to lung and neuro protection. We have not discussed caffeine for apnea of prematurity. However, we have included an expanded literature related to animal/human research to explain caffeine actions.

In the abstract – caffeine is described as a psychoactive drug – perhaps not the best description in the first few sentences

We thank the reviewer for the suggestion and appropriate changes made in the abstract. The sentence is modified. Caffeine, is the most widely used drug by both adults and children worldwide for its ability to promote alertness and elevating mood. It is effective in the management of apnea of prematurity in premature infants.

It would be important to describe the most recent results of the CAP trial

We have cited almost all the CAP trial related studies until 2018.

The abstract should be modified to take into consideration the pre-natal animal results.

We have made appropriate changes to the abstract. However, in animal models, exposure to caffeine during pregnancy and lactation adversely affects neuronal development and adult behavior of their offspring. Prenatal caffeine predisposes to intrauterine growth restriction and small for gestational age at birth. However, in-utero exposure to caffeine is also associated with excess growth, obesity and cardio-metabolic changes in children.

Last sentence of the abstract - The long-term consequences of caffeine in adults born preterm are being studied and may depend on the ability of caffeine to modulate both the expression and the maturation of adenosine receptors in infants treated with caffeine.   

The reference list is incomplete.

We have extensively revised the references. The number of references has increased from 110 to 130.